# Integrating Solving Forward and Inverse Problems in PDEs with Flow-based Models

## Abstract

Solving partial differential equations (PDEs) given input parameters (forward problem) and inferring unknown parameters from partially observed solutions (inverse problem) are two critical problems in scientific computing. Most existing approaches treat forward problems and inverse problems as separate ones. In this article, we present a novel method based on rectified flow that integrates the solution of both problems in a single model. Specifically, in the training stage, we approximate the velocity field in rectified flow with fixed pairs $z(0) = a$ and $z(T) = u$, where $a$ is the input parameter and $u$ is the corresponding solution. In the inference stage, the forward problem can be solved by feeding $z(0)$ with $a$ and running the forward pass of the flow model, and the inverse problem can be solved by feeding $z(T)$ with $u$ and running the reverse pass of the flow model. Numerical results on various equations demonstrate that the proposed method achieves competitive accuracy in solving forward problems and shows better performance in solving inverse problems within a single model.

## 1 Introduction

Partial differential equations (PDEs) are fundamental to modeling complex phenomena in physics, engineering, and beyond. Solving both forward problems (predicting solutions from parameters) and inverse problems (inferring parameters from solutions) is critical in applications ranging from fluid dynamics to geophysical exploration. Traditional numerical methods, such as finite elements or finite differences, provide rigorous solutions but suffer from high computational costs, especially for large-scale or geometrically complex problems.

Neural operators, such as DeepONet Lu et al. (2021) and Fourier Neural Operator Li et al. (2021) (FNO), offer a data-driven alternative by learning solution operators from data, thereby bypassing mesh generation and discretization. However, these approaches typically specialize in either forward or inverse problems, requiring task-specific designs or retraining. This separation not only increases development overhead but also introduces error accumulation and limits generalization.

Rectified flow Liu et al. (2023), a continuous generative model, offers a promising direction due to its inherent reversibility: by learning a continuous velocity field, it supports both forward integration (from initial to target state) and reverse integration (from target to initial state). This property naturally aligns with the bidirectional nature of PDE solving, yet remains unexplored in this context.

To address this gap, we propose the *Rectified Flow-Based Operator* (RFO), which unifies forward and inverse PDE solving within a single model based on rectified flow. In this method, the velocity field is trained on the forward problems, and we run the forward pass of the model to solve the forward problems. For the inverse problems, we simply run the reverse process of the model to approximate the unknown parameters. Therefore, we can integrate both problems in a single model based on rectified flow.

We summarize the contributions of this study in the following:

- We reformulate the forward problem in PDEs as a flow-based generative model, and present a novel method that trains the velocity field to solve the forward problems in PDEs.

- The proposed method directly utilizes the reverse process of the flow-based model that is pre-trained on the forward problem, and thus requires no extra training steps for solving inverse problems.

- Numerical results show that the proposed method outperforms Fourier Neural Operator both in accuracy and efficiency in forward problems, and that the proposed method achieves comparable results in inverse problems in a zero-shot regime.

## 2 RELATED WORK

We present the related work on forward and inverse problems in PDEs in the following.

### 2.1 NEURAL OPERATORS FOR FORWARD PROBLEMS

Neural operators represent a paradigm shift in PDE solving by learning the solution operator directly from data. DeepONet Lu et al. (2021) pioneered this approach by decomposing the input space into branch and trunk networks, learning mappings of the form: where the branch net processes input parameters and the trunk net handles spatial coordinates. While effective for many problems, its reliance on local feature aggregation limits performance in problems with strong global dependencies. Fourier Neural Operator (FNO) Li et al. (2021) addresses this limitation by using global integral operators implemented by Fourier layers, where each Fourier layer mixes frequency and spatial features. FNO achieves exceptional accuracy in forward problems by capturing long-range dependencies, outperforming DeepONet in fluids and porous media flows. GNO Anandkumar et al. (2019), MGNO Li et al. (2020) and GINO Li et al. (2023) utilize graph data to solve forward problems in irregular geometries. GNOT Hao et al. (2023) applies Transformer Vaswani et al. (2017) as the base model to achieve better performances in real-world applications. PINO Li et al. (2024) proposed to utilize physical information to improve the performance.

### 2.2 NUMERICAL SOLVERS FOR INVERSE PROBLEMS

Inverse problems infer the input parameters from the partially observed data, and has been widely applied in various engineering applications, e.g., oil exploration, airfoil design, and so on. Traditional approaches include Tikhonov regularization, Bayesian inference, and adjoint-based methods. In the neural operator literature, most approaches involve either: (1) Direct inversion: Training a separate network to map solutions to parameters, which often requires large datasets and may not generalize well; and (2) Iterative optimization: Using the forward operator within an optimization loop to minimize the data misfit, which can be computationally expensive. Despite the advances in traditional methods, the challenges of handling high-dimensional problems and complicated geometries still exist in practice. Based on the advancements of deep learning, various methods Jiang et al. (2025); Wang & Wang (2024); Molinaro et al. (2023); Wang et al. (2025) are proposed to approximate the solutions of inverse problems, and have shown impressive results.

Recent advances in generative models Goodfellow et al. (2014); Ho et al. (2020); Liu et al. (2023) provide new insights for modeling forward and inverse problems in PDEs. Generative models learn the map between known prior distribution and unknown data distribution in probability space, thus can be generalized to learning operators in function space. Wang & Wang (2024); Huang et al. (2024) utilize the diffusion model to map the noise to the solutions of forward or inverse problems, and thus solve the problems in PDEs.

Existing methods fail to unify forward and inverse solving in a single framework: FNO excels in forward problems but struggles with inverse tasks; diffusion models handle noisy inverses but lack forward precision and require stochastic sampling. This divide necessitates a framework that supports both directions within a single model while maintaining computational efficiency and accuracy.

## 3 PRELIMINARIES

### 3.1 FORWARD AND INVERSE PROBLEMS IN PDES

We consider the following general formulation of partial differential equation:

$$\begin{cases} \mathcal{N}(u; a) = 0, & x \in \Omega, \\ \mathcal{B}u = g, & x \in \partial\Omega \end{cases} \tag{1}$$

where $u : \mathcal{X} \to \mathbb{R}^{d_u}$ is the unknown function, $a : \mathcal{X} \to \mathbb{R}^{d_a}$ is the input parameter, $\mathcal{N}$ is the differential operator, $\mathcal{B}$ is the boundary operator (Dirichlet-, Neumann-, and Robin-type), and $g$ is the boundary condition. The input parameter $a$ can be the initial condition in time-dependent problems, the coefficient function in diffusion equations, and the source term in wave equations.

Forward problem learns the map from the input parameter to the solution, i.e.,

$$\mathcal{G} : \mathcal{A} \to \mathcal{U},$$

where $\mathcal{A}$ and $\mathcal{U}$ are the Banach spaces that $a$ and $u$ lie in, respectively. Learning the map $\mathcal{G}$ is also well-known as operator learning, while we use the terminology "forward problem" for consistency in illustration. Various problems in scientific computing can be modeled as forward problems, such as velocity prediction in fluids, weather forecasting, and so on.

In many engineering scenarios, people wish to infer some important unknown coefficient from the partially observed data $u \in \mathcal{U}$, which can be formulated as the inverse problem that maps from the solution to the input parameter in the following:

$$\mathcal{G}^{-1} : \mathcal{U} \to \mathcal{A}.$$

For example, we infer the initial condition $a \in \mathcal{A}$ from the partially observed data $u \in \mathcal{U}$ at $t = T$ in time-dependent problems, the conductivity coefficient $a \in \mathcal{A}$ in diffusion equations, and the source term $a \in \mathcal{A}$ in wave equations. Inverse problem has broadly impact in engineering applications, which are widely applied in oil exploration, airfoil design.

### 3.2 RECTIFIED FLOW

Rectified flow Liu et al. (2023) is a generative framework that learns to straighten the probability flow between two distributions. Given a source distribution $p_0$ (parameters) and target distribution $p_1$ (solutions), rectified flow defines a straight-line ODE path:

$$\frac{d\mathbf{z}_t}{dt} = \mathbf{v}(\mathbf{z}_t, t), \quad \mathbf{z}_0 \sim p_0, \quad t \in [0, 1],$$

with the linear interpolation path given by $\mathbf{z}_t = \mathbf{z}_0 + t(\mathbf{z}_1 - \mathbf{z}_0)$. The velocity field $\mathbf{v}_\theta$ is parameterized by a neural network and trained to minimize the following loss:

$$\mathcal{L}(\theta) = \mathbb{E}_{(\mathbf{z}_0, \mathbf{z}_1) \sim \Pi(p_0, p_1), t \sim \mathcal{U}(0,1)} \left[ \|\mathbf{v}_\theta(\mathbf{z}_t, t) - (\mathbf{z}_1 - \mathbf{z}_0)\|^2 \right],$$

where $\Pi(p_0, p_1)$ is the joint distribution with marginal densities $p_0$ and $p_1$, respectively, $\mathbf{z}_t = \mathbf{z}_0 + t(\mathbf{z}_1 - \mathbf{z}_0)$. Through iterative training and resampling, the paths become progressively straighter, enabling efficient sampling with fewer steps.

## 4 RECTIFIED FLOW-BASED OPERATOR

The key innovation of RFO lies in modeling the parameter-and-solution relationship of PDEs as a deterministic continuous evolution process using rectified flow. Unlike conventional methods that treat forward and inverse tasks independently, RFO leverages the reversibility of rectified flow—the same velocity field drives both the "parameter-to-solution" forward evolution and the "solution-to-parameter" reverse evolution. This design eliminates the need for retraining or modifying the model architecture when switching between tasks.

## 4.1 MATHEMATICAL FOUNDATION OF RFO

### 4.1.1 RECTIFIED FLOW MODELING FOR PDEs

For any target PDE, we define two core components:

1. PDE Parameters ($a$): initial condition $u_0(x)$ for Burgers' equation, permeability $a(\mathbf{x})$ for Darcy flow;

2. PDE Solutions ($u$): scalar field $u(x, 1)$ at $t = 1$ for Burgers' equation, pressure field $u(\mathbf{x})$ for Darcy flow.

RFO reformulates the parameter-and-solution mapping as a noise-free ordinary differential equation (ODE). This ODE describes how the state evolves from the PDE parameter $a$ to the solution $u$ over the normalized time interval $t \in [0, 1]$:

$$\frac{\mathrm{d}\mathbf{z}_t}{\mathrm{d}t} = \mathbf{v}_\theta(\mathbf{z}_t, t), \quad \mathbf{z}_0 = \mathbf{a}.$$

where: $\mathbf{z}_0 = a$: Initial state at $t = 0$, corresponding to the input PDE parameter; $\mathbf{z}_1 = u$: Final state at $t = 1$, corresponding to the output PDE solution; $\mathbf{z}_t$: The state at time $t$, which directly represents either PDE parameters (at $t = 0$) or solutions (at $t = 1$); $\mathbf{v}_\theta(\mathbf{z}_t, t)$: A continuous velocity field parameterized by a neural network (with trainable parameters $\theta$), learned to drive the state transition from $a$ to $u$.

**Critical Reversibility Property**   The noise-free nature of the ODE guarantees reversible evolution. To solve the inverse problem (inferring $a$ from $u$), we simply reverse the time direction of the ODE:

$$\frac{\mathrm{d}\mathbf{y}_\tau}{\mathrm{d}\tau} = -\mathbf{v}_\theta(\mathbf{y}_\tau, 1 - \tau), \quad \mathbf{y}_0 = \mathbf{u}.$$

This reversibility ensures no additional model is required for inverse tasks—only the evolution direction of the pre-trained velocity field $v_\theta$ is adjusted.

### 4.1.2 TRAINING OF THE VELOCITY FIELD

The velocity field $\mathbf{v}_\theta$ is trained to minimize the trajectory alignment loss, which ensures the ODE accurately maps parameters to solutions. The loss function is defined as:

$$\mathcal{L}(\theta) = \mathbb{E}_{a,u,t} \left\| \mathbf{v}_\theta(\mathbf{z}_t, t) - (\mathbf{u} - \mathbf{a}) \right\|^2,$$

where: $(a, u)$: Paired PDE parameter-and-solution samples (e.g., $a = u_0(x)$ and $u = u(x, 1)$ for Burgers' equation) in the training dataset; $t$: random time sampled from $[0, 1]$, used to generate intermediate states for alignment; $z_t = a + t \cdot (u - a)$: the linear interpolation between $a$ and $u$, serving as the target intermediate state to guide velocity field learning.

---

**Algorithm 1** Training process of Rectified-Flow based Operator.

---

1: Generate paired datasets $\{(a_i, u_i)\}$ for the target PDE (e.g., 10,000 pairs for Darcy flow).
2: **repeat**
3:    For each pair $(a_i, u_i)$, sample a random time $t \sim \text{Uniform}([0, 1])$ and compute the intermediate state $z_{t,i} = a_i + t \cdot (u_i - a_i)$.
4:    Compute the loss $\mathcal{L}(\theta) = \mathbb{E}_{a_i, u_i, t} \left\| v_\theta(z_{t,i}, t) - (u_i - a_i) \right\|^2$.
5:    Take a gradient descent step on $\nabla_\theta \mathcal{L}(\theta)$ using the Adam optimizer (learning rate = $10^{-4}$).
6: **until** loss converges (typically 500–1000 epochs)

---

## 4.2 INFERENCE WITH RECTIFIED FLOW-BASED OPERATOR

### 4.2.1 FORWARD PROBLEM

Given a PDE parameter $a$ (e.g., initial condition $u_0(x)$), the solution $u$ is obtained via forward ODE integration with a preset time step $\Delta t$ (e.g., $\Delta t = 0.1$, total steps $N = 10$):

Figure 1: RFO Workflow

1. Initialize the state: $z_0 = a$.

2. Iterate over time steps $t = 0$ to $N - 1$:

$$z_{t+\Delta t} = z_t + \Delta t \cdot v_\theta(z_t, t \cdot \Delta t)$$

3. Output the final state as the PDE solution: $u = z_1$.

### 4.2.2 INVERSE PROBLEM

Given a PDE solution $u$ (e.g., pressure field $u(\mathbf{x})$), the parameter $a$ is inferred via reverse ODE integration (using the same preset $\Delta t$):

1. Initialize the state: $z_1 = u$.

2. Iterate over time steps $t = N - 1$ to $0$:

$$z_{t \cdot \Delta t} = z_{(t+1) \cdot \Delta t} - \Delta t \cdot v_\theta(z_{(t+1) \cdot \Delta t}, (t+1) \cdot \Delta t)$$

3. Output the initial state as the PDE parameter: $a = z_0$.

## 5 EXPERIMENTS

To verify the effectiveness of the proposed RFO framework in unifying forward and inverse PDE problems, we conduct experiments on four classic PDEs (Burgers' equation, Darcy flow, Navier-Stokes equations, and 1D advection equation). For each problem, we first clarify the definition of forward and inverse tasks, then analyze the experimental results to demonstrate the framework's performance.

### 5.1 BURGERS' EQUATION

Burgers' equation is a typical nonlinear PDE that exhibits shock wave formation, making it ideal for testing the framework's ability to capture sharp spatial changes.

#### 5.1.1 TASK DEFINITION

**Forward Problem**: Given the initial condition $u_0(x) \in L^2_{per}((0, 1); \mathbb{R})$ (scalar field distribution at $t = 0$) and viscosity coefficient $\nu \in \mathbb{R}_+$, solve for the scalar field $u(x, t)$ at the final time $t = 1$. The governing equation and boundary/initial conditions are:

$$\begin{cases} \frac{\partial u(x,t)}{\partial t} + u(x,t)\frac{\partial u(x,t)}{\partial x} = \nu \frac{\partial^2 u(x,t)}{\partial x^2}, & (x,t) \in (0,1) \times (0,1] \\ u(x,0) = u_0(x), & x \in (0,1) \\ u(0,t) = u(1,t), & t \in [0,1] \end{cases}$$

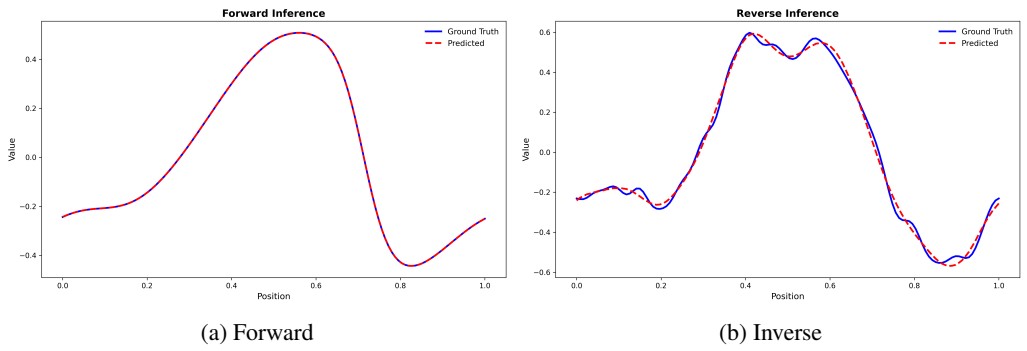

(a) Forward             (b) Inverse

Figure 2: Numerical results for Burgers' equation with resolution 1024.

where the periodic boundary condition $u(0, t) = u(1, t)$ ensures no reflection of the scalar field at the domain boundary.

**Inverse Problem**: Given the scalar field $u(x, 1)$ (solution at $t = 1$) and viscosity coefficient $\nu$, infer the initial condition $u_0(x)$ at $t = 0$. This task requires reversing the "time evolution" mapping of the forward problem, which is challenging due to the nonlinearity-induced information compression near shock waves.

### 5.1.2 EXPERIMENTAL RESULTS

RFO achieves superior accuracy in forward solving and robust inverse inference compared to baselines (GNO, LNO, FNO). As shown in Table 1, RFO's forward $L^2$ error (e.g., 0.0016 at $s = 256$) is an order of magnitude lower than FNO's (e.g., 0.0149 at $s = 256$); in inverse tasks, RFO's error (e.g., 0.0623 at $s = 256$) outperforms FNO's (e.g., 0.0751 at $s = 256$).

Figure 2a illustrates that RFO's forward prediction (blue curve) closely overlaps with the ground truth (black curve), even near shock wave regions; Figure 2b shows RFO's inverse inference (red curve) accurately reconstructs the initial condition from the final solution, verifying its shock-capturing and reverse mapping capabilities.

Table 1: Numerical results on 1D Burgers' equation.

| Task | Method | Resolution | | | | | |
|------|--------|-----------|-----------|------------|------------|------------|------------|
| | | $s = 256$ | $s = 512$ | $s = 1024$ | $s = 2048$ | $s = 4096$ | $s = 8192$ |
| Forward | GNO | 0.0555 | 0.0594 | 0.0651 | 0.0663 | 0.0666 | 0.0699 |
| | LNO | 0.0212 | 0.0221 | 0.0217 | 0.0219 | 0.0200 | 0.0189 |
| | FNO | 0.0149 | 0.0158 | 0.0160 | 0.0146 | 0.0142 | 0.0139 |
| | RFO (Ours) | **0.0016** | **0.0017** | **0.0015** | **0.0015** | **0.0017** | **0.0016** |
| Inverse | FNO | 0.0751 | 0.0753 | 0.0744 | 0.0752 | 0.0772 | 0.0743 |
| | RFO (Ours) | **0.0623** | **0.0621** | **0.0607** | **0.0640** | **0.0624** | **0.0632** |

## 5.2 DARCY FLOW

Darcy flow describes fluid flow in porous media, characterized by a second-order elliptic PDE with discontinuous diffusion coefficients (permeability), testing the framework's ability to handle parameter inhomogeneity.

### 5.2.1 TASK DEFINITION

**Forward Problem**: Given the diffusion coefficient $a(\mathbf{x}) \in L^\infty((0, 1)^2; \mathbb{R}_+)$ (permeability distribution of porous media) and forcing function $f(\mathbf{x}) \in L^2((0, 1)^2; \mathbb{R})$ (source/sink term), solve for the

pressure field $u(\mathbf{x})$ (solution of the PDE). The governing equation and boundary conditions are:

$$\begin{cases} -\nabla \cdot (a(\mathbf{x})\nabla u(\mathbf{x})) = f(\mathbf{x}), & \mathbf{x} \in (0,1)^2 \\ u(\mathbf{x}) = 0, & \mathbf{x} \in \partial(0,1)^2 \end{cases}$$

where the Dirichlet boundary condition $u(\mathbf{x}) = 0$ represents zero pressure at the unit box boundary.

**Inverse Problem**: Given the pressure field $u(\mathbf{x})$ (solution) and forcing function $f(\mathbf{x})$, infer the diffusion coefficient $a(\mathbf{x})$ (permeability). This task is challenging due to the discontinuity of $a(\mathbf{x})$ and the ill-posedness of elliptic inverse problems (small changes in $u(\mathbf{x})$ may cause large deviations in $a(\mathbf{x})$).

### 5.2.2 EXPERIMENTAL RESULTS

RFO matches FNO's forward accuracy and provides more stable inverse solutions. As shown in Table 2, RFO's forward error (e.g., 0.0108 at $s = 85$) is comparable to FNO's (e.g., 0.0108 at $s = 85$), and even outperforms FNO at higher resolutions (e.g., 0.0092 vs. 0.0098 at $s = 421$). In inverse tasks, RFO maintains consistent performance across resolutions (e.g., 0.1315 at $s = 85$, 0.1125 at $s = 421$), while baselines like GNO and LNO either fail to handle high resolutions (LNO has no result at $s = 421$) or show higher errors.

Figure 3a demonstrates that RFO accurately predicts the pressure field from the permeability distribution, capturing the pressure drop near high-permeability regions; Figure 3b shows RFO reconstructs the discontinuous permeability field from pressure data, verifying its ability to handle inhomogeneous parameters.

Table 2: Numerical results on 2D Darcy flow.

| Task | Method | \multicolumn{4}{c}{Resolution} | | | |
|------|--------|--------|---------|---------|---------|
| | | $s = 85$ | $s = 141$ | $s = 211$ | $s = 421$ |
| Forward | GNO | 0.0346 | 0.0332 | 0.0342 | 0.0369 |
| | LNO | 0.0520 | 0.0461 | 0.0445 | – |
| | MGNO | 0.0416 | 0.0428 | 0.0428 | 0.0420 |
| | FNO | 0.0108 | 0.0109 | 0.0109 | **0.0098** |
| | RFO (Ours) | **0.0108** | **0.0108** | **0.0101** | 0.0101 |
| Inverse | FNO | 0.1521 | 0.1487 | 0.1453 | 0.1392 |
| | RFO (Ours) | **0.1315** | **0.1324** | **0.1317** | **0.1325** |

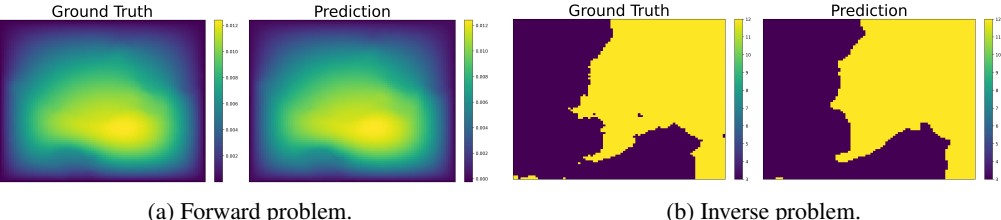

        (a) Forward problem.               (b) Inverse problem.

Figure 3: Numerical results on Darcy flow with resolution $s = 211$.

### 5.3 NAVIER-STOKES EQUATIONS

The Navier-Stokes equations describe viscous incompressible fluid flow, featuring complex turbulent dynamics, testing the framework's ability to model time-dependent nonlinear transport.

### 5.3.1 TASK DEFINITION

We use the vorticity formulation to simplify the equations, where $w(\mathbf{x}, t) = \nabla \times \mathbf{u}(\mathbf{x}, t)$ is the vorticity (scalar field) and $\mathbf{u}(\mathbf{x}, t)$ is the velocity field (satisfying incompressibility $\nabla \cdot \mathbf{u} = 0$).

**Forward Problem**: Given the initial vorticity $w_0(\mathbf{x}) \in L^2_{per}((0,1)^2; \mathbb{R})$ (at $t = 0$), viscosity coefficient $\nu \in \mathbb{R}_+$, and forcing function $f(\mathbf{x}) \in L^2_{per}((0,1)^2; \mathbb{R})$, predict the vorticity $w(\mathbf{x},t)$ for $t \in (10, T]$ (focusing on late-time turbulent evolution). The governing equations are:

$$\begin{cases} \frac{\partial w(\mathbf{x},t)}{\partial t} + \mathbf{u}(\mathbf{x},t) \cdot \nabla w(\mathbf{x},t) = \nu \Delta w(\mathbf{x},t) + f(\mathbf{x}), & (\mathbf{x},t) \in (0,1)^2 \times (0,T] \\ \nabla \cdot \mathbf{u}(\mathbf{x},t) = 0, & (\mathbf{x},t) \in (0,1)^2 \times [0,T] \\ w(\mathbf{x},0) = w_0(\mathbf{x}), & \mathbf{x} \in (0,1)^2 \end{cases}$$

where $T = 50$ (total simulation time) and the periodic boundary condition is implied by $L^2_{per}$ space.

**Inverse Problem**: Given the vorticity sequence $\{w(\mathbf{x},t)\}_{t \in (10,T]}$ (observed late-time turbulent data), infer the vorticity $w(\mathbf{x}, 10)$ at $t = 10$ (key intermediate state for flow evolution). This task requires reversing the turbulent time evolution, which is challenging due to the chaotic nature of turbulence.

### 5.3.2 EXPERIMENTAL RESULTS

RFO achieves competitive performance in both forward and inverse tasks with improved efficiency. As shown in Table 3, RFO has fewer parameters (5.5M) than FNO-3D (6.6M) and TF-Net (7.5M), while its training time per epoch (14.24s) is significantly lower than FNO-2D (127.80s) and ResNet (78.47s). In forward tasks, RFO achieves the best error at $\nu = 10^{-4}$ (0.1138) and $\nu = 10^{-5}$ (0.2049); in inverse tasks, RFO's error (e.g., 0.0514 at $\nu = 10^{-3}$) outperforms most baselines, verifying its ability to model turbulent dynamics and reverse time evolution. Visual results are presented in Table 4, and show the effectiveness of the proposed method.

Table 3: Performance comparison on Navier-Stokes equations with resolution $64 \times 64$.

| Task | Method | Parameters | Time/epoch (s) | Viscosity | | |
| --- | --- | --- | --- | --- | --- | --- |
| | | | | $\nu = 10^{-3}$ | $\nu = 10^{-4}$ | $\nu = 10^{-5}$ |
| Forward | ResNet | 266,641 | 78.47 | 0.0701 | 0.2871 | 0.2753 |
| | FNO-3D | 6,558,537 | 38.99 | **0.0086** | 0.1918 | **0.1893** |
| | FNO-2D | 414,517 | 127.80 | 0.0128 | 0.1559 | 0.1556 |
| | RFO (Ours) | 5,532,509 | 14.24 | 0.0161 | **0.1138** | 0.2049 |
| Inverse | FNO-3D | 6,558,537 | 39.46 | **0.0549** | 0.2036 | **0.4641** |
| | RFO (Ours) | 5,532,509 | – | 0.0614 | **0.1961** | 0.4857 |

## 5.4 WELL-POSED INVERSE PROBLEMS

To isolate the impact of inverse problem ill-posedness on performance, we test on the 1D Advection equation—its inverse problem is theoretically well-posed (unique and stable solution), allowing us to verify RFO's bidirectional consistency.

Table 4: Visualization results on Navier-Stokes equation.

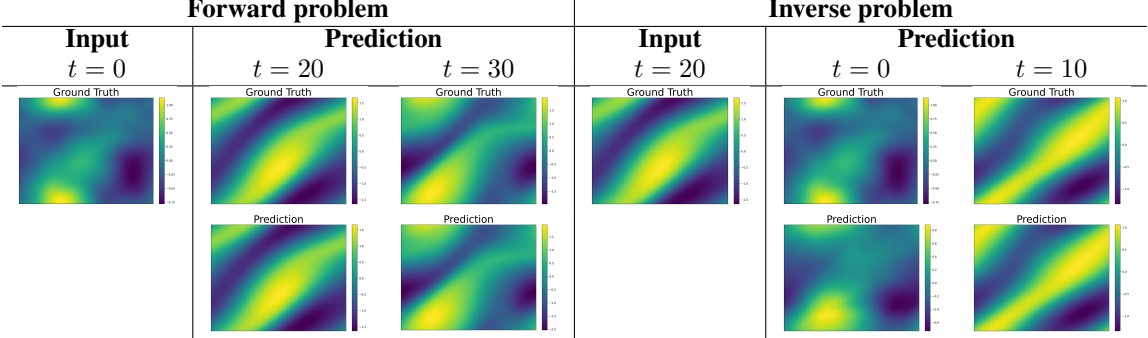

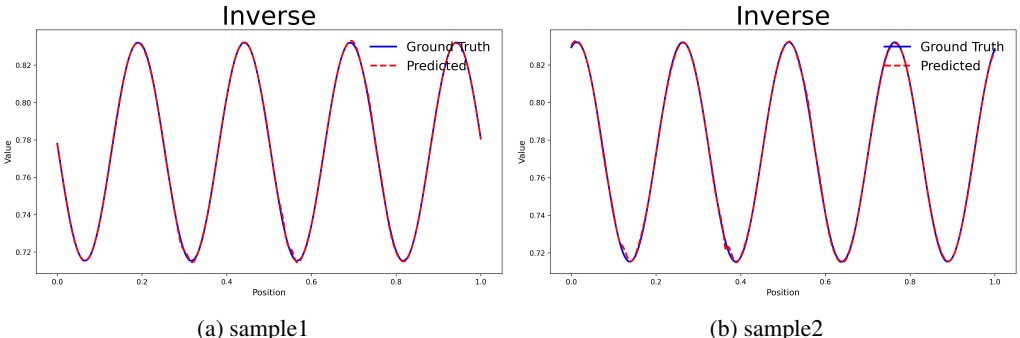

(a) sample1                  (b) sample2

Figure 4: Inverse results for 1D advection equation.

### 5.4.1 TASK DEFINITION

The Advection equation describes the translation of a scalar field (e.g., concentration, temperature) in a uniform flow, with no diffusion or nonlinearity, ensuring no information loss during evolution.

**Forward Problem**: Given the initial scalar field $u_0(x) \in L^2_{per}((0,1); \mathbb{R})$ (at $t = 0$) and constant advection velocity $c \in \mathbb{R}_+$, solve for the scalar field $u(x,t)$ at $t = T = 1$. The governing equation and boundary/initial conditions are:

$$\begin{cases} \frac{\partial u(x,t)}{\partial t} + c\frac{\partial u(x,t)}{\partial x} = 0, & (x,t) \in (0,1) \times (0,1] \\ u(x,0) = u_0(x), & x \in (0,1) \\ u(0,t) = u(1,t), & t \in [0,1] \end{cases}$$

The analytical solution is $u(x,t) = u_0(x - ct \mod 1)$, meaning the scalar field translates uniformly without distortion.

**Inverse Problem**: Given the scalar field $u(x,1)$ (at $t = 1$) and advection velocity $c$, infer the initial scalar field $u_0(x)$. The analytical inverse solution is $u_0(x) = u(x + ct \mod 1)$—the inverse task is simply reversing the translation direction, with no ill-posedness (unique solution, no sensitivity to noise).

### 5.4.2 EXPERIMENTAL RESULTS

RFO demonstrates balanced accuracy in both forward (predicting $u(x,1)$ from $u_0(x)$) and inverse (inferring $u_0(x)$ from $u(x,1)$) tasks on the 1D advection equation, verifying the theoretical bidirectional consistency. Numerical results in Table 5 shows the performance gain under different levels of noise compared to FNO which requires training on the reversed data pair $(u, a)$, which verifies the effectiveness of RFO in zero-shot inverse problems. Figure 4 presents the visual performance of the proposed method.

Table 5: L2 relative error ($\times 10^{-2}$) on the inverse problems under Gaussian noise.

| Method | Noise level | | | | |
| --- | --- | --- | --- | --- | --- |
| | $\sigma = 0.01$ | $\sigma = 0.05$ | $\sigma = 0.10$ | $\sigma = 0.20$ | $\sigma = 0.30$ |
| FNO | 0.1745 | **0.2874** | 0.7184 | 1.5974 | 2.3320 |
| RFO (Ours) | **0.1072** | 0.2937 | **0.5592** | **1.0987** | **1.6660** |

## 6 CONCLUSION

We presented Rectified Flow-based Neural Operator (RFO), a unified framework for solving forward and inverse problems in scientific computing. By learning the velocity field in the rectified flow

model, RFO solves the forward and inverse problems by the forward and reverse passes with a single model. For forward problems, RFO outperforms other methods in Burgers equation and Darcy flow equation, and in inverse problems, RFO outperforms the specialized method without any extra training. Future work will extend RFO to stochastic settings and more complex PDE systems.

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
