# OpenReview forum: "Integrating Solving Forward and Inverse Problems in PDEs with Flow-based Models"
_ICLR.cc/2026/Conference — ICLR 2026 Conference Withdrawn Submission_

### Official Review · Reviewer_1un3 · 2025-10-20

**Soundness:** 2
**Presentation:** 2
**Contribution:** 1
**Rating:** 0
**Confidence:** 4

**Summary:**

This paper proposes a Rectified Flow-Based Operator (RFO) that unifies the solution of forward and inverse PDE problems within a single model. By training a rectified flow velocity field between PDE parameters (inputs) and solutions (outputs), the model can perform forward prediction via forward integration and inverse inference via reverse integration, without retraining. Experiments on Burgers’, Darcy, Navier–Stokes, and Advection equations demonstrate that RFO achieves accuracy comparable to or better than existing neural operators (e.g., FNO), while also handling inverse problems in a zero-shot fashion. The paper’s main claim is that rectified flow’s reversibility naturally supports both forward and inverse tasks in a unified framework.

**Strengths:**

- The idea of using rectified flow’s reversibility to connect forward and inverse PDE tasks is novel and intuitive, bridging generative modeling and operator learning.
- The method does not require retraining or architectural changes to switch between forward and inverse problems.
- Across multiple PDE benchmarks, RFO shows accuracy comparable to or better than FNO, with improved computational efficiency in some tasks.

**Weaknesses:**

- The experiments are mostly qualitative and small-scale, lacking systematic ablation studies
- The baselines (e.g., GNO, LNO, FNO) are not fairly tuned or described, and no analysis is given for computational cost vs. accuracy trade-offs.
- The comparisons rely mostly on older operator-learning methods such as GNO, LNO, and FNO, without inclusion of recent state-of-the-art frameworks, for instance, diffusion-based PDE solvers (DiffusionPDE, Physics-Informed Diffusion Models), or PINO variants.
- The baseline LNO is not referenced
- This paper is purely experimental, but no implementation is provided, making it hard to assess the reproducibility.

**Questions:**

- Could you clarify whether RFO offers any theoretical justification (e.g., existence or uniqueness of the rectified flow mapping for PDE operators) beyond empirical validation?
- The baselines are mostly older neural operator models (GNO, FNO, LNO).
- Why were recent diffusion- or flow-based PDE solvers (e.g., DiffusionPDE (2025), PIDM (2025), FunDiff (2025)) not included?
- Could you provide more details on the network architecture (depth, hidden width, total parameters) and training hyperparameters used in each PDE task?

---

### Official Review · Reviewer_eeTH · 2025-10-30

**Soundness:** 3
**Presentation:** 3
**Contribution:** 3
**Rating:** 6
**Confidence:** 4

**Summary:**

This paper presents a novel method based on rectified flow that integrates the solution of both forward problems and inverse problems in a single model.  In the training stage, the velocity field in rectified flow is approximated  with fixed pairs z(0)=a and z(T)=u, where a is the input parameter and u is the corresponding solution.  In the inference stage, the forward problem can be solved by feeding z(0) with a and running the forward pass of the flow model, and the inverse problem can be solved by feeding z(T) with u and running the reverse pass of the flow model. Numerical results on various equations demonstrate its effectiveness in both forward problems and inverse problems within a single model.

**Strengths:**

1. Easy to follow and easy to implement.
2. Both forward problems and inverse problems are solved in a single model.
3. To my knowledge, this is the first work integrating both forward problems and inverse problems in a single model based on rectified flow.
4. The inferences in both forward problems and inverse problems are fast without extra training.

**Weaknesses:**

1. The compared baselines are mainly FNO and its variants, and there is only one baseline for inverse problems. It is suggested to compare with more recent neural operator methods, such as Neural Inverse Operators (NIOs) for inverse problems.

 Neural Inverse Operators:  Roberto Molinaro, Yunan Yang, Bj¨orn Engquist, and Siddhartha Mishra. Neural inverse operators
 for solving pde inverse problems. arXiv preprint arXiv:2301.11167,2023.

2. The training details are not given.

**Questions:**

For the initial conditions of each equation, what distributions did you use when drawing samples? And how many training and testing samples did you use?

---

### Official Review · Reviewer_4fUV · 2025-10-30

**Soundness:** 2
**Presentation:** 2
**Contribution:** 2
**Rating:** 2
**Confidence:** 4

**Summary:**

This paper presents an application of flow matching to PDEs. By learning the solution of a PDE with a flow matching approach, the method allows solving the inverse problem out-of-the-box by reversing the direction of the rectified flow from the solution to the inputs. The method is evaluated on multiple PDEs and compared against neural operators such as FNO and LNO.

**Strengths:**

- Conceptually interesting application of flow matching to inverse problems.
- The paper is generally easy to follow.
- Evaluation on a wide range of PDEs.

**Weaknesses:**

1. Only applicable to cases where the input parameters have the same dimension as the outputs.
2. For the inverse problem, the paper only compares against FNO, even when other models designed for inverse problems were referred to in the related works section. Additionally,  LNO was also used for inverse design in the original paper.
3. Formatting issues (exceeds page limit, no parentheses around citations).
4. Almost no information on the model architecture is provided. Additionally, more details on the experiments should be provided, such as the range of the input parameters, the number of training trajectories, and how the baselines were applied.

**Questions:**

1. How did you solve the inverse problem with FNO?
2. For the different resolutions, did you repeat the training for each resolution level, or does this refer to a super-resolution setting?

---

### Official Review · Reviewer_qcwW · 2025-11-01

**Soundness:** 2
**Presentation:** 3
**Contribution:** 2
**Rating:** 4
**Confidence:** 4

**Summary:**

This paper leverages the Rectified Flow framework introduced by Liu et al. (2023) to develop a unified approach for solving both forward and inverse PDE problems with a single trained model. The key idea is to treat PDE parameters and solutions as the initial and terminal states of a rectified flow, respectively. A neural network parameterizes the velocity field governing the flow, and the model is trained by minimizing a trajectory alignment loss. Once trained on forward-problem data, the model can solve inverse problems by reversing the ODE dynamics, without requiring retraining. The paper includes empirical evaluations demonstrating the effectiveness of the approach relative to existing methods.

**Strengths:**

The idea of unifying forward and inverse PDE solving within a single framework is novel and conceptually appealing. Leveraging Rectified Flow to model the transition between PDE parameters and solutions is a reasonable.

**Weaknesses:**

1.  The text states in line 146 that the velocity field v_\theta is parameterized by a neural network, but the architecture and configuration of the network are not specified.

2. Insufficient presentation of experimental results. In the experiments, RFO is compared with multiple baseline methods, including GNO, LNO, and FNO for forward problems, whereas for inverse problems, it is only compared with FNO. In Wang & Wang (2024), experimental results of LNO on the 1D Burgers inverse problem are provided, whereas this paper does not present a comparison with these results. Furthermore, this paper also lacks sufficient comparison with existing models, such as GNOT by Hao et al. (2023) and Transolver by Wu et al. (2024).

3. Lacks further theoretical explanation/justification. Rectified Flow aims to construct a transport path between the source and target distributions that is as “straight” as possible, including rectification and recursive refinement steps. The paper does not provide a theoretical justification for why using Rectified Flow to model the relationship between PDE parameters and solutions is reasonable. This lack of explanation leaves me somewhat confused about the theoretical soundness of the proposed approach.

**Questions:**

1. Clarify the architecture of the neural network modeling the velocity field (e.g., network type, depth, width, activation functions, etc.).
2. In the experiments, LNO is included as a baseline for forward problems but not for inverse problems. Could the authors clarify the reason for omitting LNO in the inverse problem comparison?
3. Several advanced models are not included in the experimental comparisons, such as GNOT (Hao et al., 2023) and Transolver (Wu et al., 2024). Could the authors comment on the rationale for not including these baselines in either forward or inverse problem evaluations?
4. It would be helpful if the authors could provide a brief theoretical explanation for why the rectification and recursive steps in Rectified Flow are suitable for modeling both forward and inverse PDE problems, without requiring extensive formal proofs.

---

### Note · Authors · 2025-12-03

I have read and agree with the venue's withdrawal policy on behalf of myself and my co-authors.